# Research on the Influence of Non-Cognitive Ability and Social Support Perception on College Students’ Entrepreneurial Intention

**DOI:** 10.3390/ijerph191911981

**Published:** 2022-09-22

**Authors:** Wentao Si, Jiayi Tian, Qi Yan, Wenshu Wang, Maocong Zhang

**Affiliations:** 1School of Public Administration, Shandong Normal University, Jinan 250014, China; siwentao@sdnu.edu.cn (W.S.); 201829010205@stu.sdnu.edu.cn (J.T.); 2021021094@stu.sdnu.edu.cn (Q.Y.); 201929010222@stu.sdnu.edu.cn (W.W.); 2Research Center for Educational Policy and Management, Shandong Normal University, Jinan 250014, China

**Keywords:** non-cognitive ability, social support perception, college students’ entrepreneurial intention, intermediary effect

## Abstract

The entrepreneurship of college students is an important issue related to the harmony and sustainable development of society as a whole. At present, the existing research in the industry pays less attention to the influence mechanism of non-cognitive ability and social support perception on college students’ entrepreneurial intention. Using 450 survey data, this paper examines the relationship between non-cognitive ability and college students’ entrepreneurial intention in terms of five dimensions: openness, conscientiousness, extraversion, agreeableness, and emotional stability. At the same time, it focuses on the role of social environmental factors, namely, social support perception in the relationship between the non-cognitive ability and entrepreneurial intention, and explores the influence path. The results show that openness, conscientiousness, extroversion, and emotional stability have significant positive effects on entrepreneurial intention; agreeableness has no significant effect on entrepreneurial intention; openness, conscientiousness, extraversion, agreeableness, and emotional stability have significant positive effects on social support perception. The mediating effect of social support perception is as follows—it is part of the intermediary effect between openness, conscientiousness, extraversion, and emotional stability on entrepreneurial intention; within the influence of agreeableness on entrepreneurial intention, it plays a complete intermediary role. This paper enriches the research results on the impact of non-cognitive ability on entrepreneurial intention, reveals the intermediary effect of social support perception on the impact of non-cognitive ability on college students’ entrepreneurial intention, and broadens the field of vision for the study of college students’ entrepreneurial intention. The research results can provide a decision-making reference for the promotion of the entrepreneurial intention of college students, alleviating the employment pressure of college graduates in China and promoting sustainable economic development.

## 1. Introduction

With the acceleration of social transformation in China, the employment pressure of college students is increasing. Entrepreneurship has gradually become an important career choice for college students and graduates. Encouraging college students’ innovation and entrepreneurship is of great significance in order to relieve the employment pressure of college students and promote economic development [1]. Therefore, to guide and encourage college students to start their own businesses, the first problem to be solved is to help college students to form their entrepreneurial intention. The entrepreneurial intention of college students is essentially an inner activity that predicts the possibility of college students starting a business in the future [2]. Therefore, entrepreneurial intention is the best predictor of entrepreneurial behavior [3]. Of course, no human action can be initiated without intending to do it. However, simply intending to become an entrepreneur does not provide a certain type of entrepreneurial behavior [4]. In this matter, clarifying the concept of entrepreneurial behavior is of utmost importance. However, a study found that entrepreneurial behavior is driven by entrepreneurial intention, and without entrepreneurial intention, there would be no entrepreneurial behavior among college students [5]. Therefore, entrepreneurial intention has gradually become a focus of researchers.

The entrepreneurial intention of college students is increasingly affected by a wide range of social backgrounds and social structures. The two most important factors in the influencing mechanism of entrepreneurial intention are the personal factors of college students and external social environment factors. First of all, from the perspective of college students’ own psychological factors, human capital is an important factor affecting their entrepreneurial intention [6]. Human capital factors such as gender, age, skills, physical fitness, knowledge, and awareness levels have a profound impact on this process [7,8]. In addition, Heckman proposed, in the new human capital theory, that non-cognitive ability is a comment on personal ability and plays an important role in personal development [9]. Previous studies have shown that non-cognitive ability can significantly improve the adaptability of workers in unstable systems and ensure that individuals still maintain the same thinking, feeling, and behavior patterns in unexpected situations [10]. At the same time, non-cognitive ability also has an important impact on workers’ income market competitiveness, career choices, and the social behavior of workers [11]. Specifically, in the aspect of college students’ entrepreneurship, non-cognitive ability affects the process of college students’ entrepreneurship to a certain extent by affecting their behavior in the face of external social aspects [12,13]. Second, college students have the personality characteristics of entrepreneurs—that is, non-cognitive ability—but they also need support from society to aid their entrepreneurial success [14]. Entrepreneurship is an economic activity; it concerns human action initiated in an economic environment based on voluntary exchange [15]. The entrepreneurial intention of college students is inevitably affected by the external social environment. As highlighted in some economic literature, college student entrepreneurship requires an appropriate institutional framework [16]. In the study of institutional framework, most scholars divide the institutional environment into formal institutions and informal institutions [17]. As two inseparable parts of the system, both systems are a unity of opposites, which are interdependent and can be transformed into each other under certain conditions [18]. Formal institutions are always associated with state power or an organization, and refer to such behavioral norms, such as various written laws, regulations, policies, rules, contracts, etc. [19]. They are identified in some definite form and are monitored and enforced by the actor’s organization. Strong formal institutions can increase the efficiency of business transactions and reduce transaction costs, thereby enabling individuals to profit from business activities [20]. A well-developed formal system increases the possibility of potential entrepreneurs to obtain business value from entrepreneurial opportunities, thus increasing their entrepreneurial willingness [21]. In environments that support entrepreneurship, regulatory-related barriers to entrepreneurship are lower, and individuals with high entrepreneurial self-efficacy are more willing to start new businesses. Informal systems refer to the unwritten restrictions on human behavior, which is a concept opposite to formal systems such as law, and include social code of conduct, social norms, moral concepts, customs, and cultural values [22]. Different from formal institutions, informal institutions are spontaneous, non-coercive, extensive, and persistent. Informal institutions indicate a state’s expected behavior and sanctions for behavior that does not adhere to social norms and values [23]. In the context of a culture that supports entrepreneurship, individuals will feel the supportive attitude of the entire society towards entrepreneurship [24]. This will stimulate individuals’ confidence in successful entrepreneurship and increase their willingness to start a business [25]. However, if a country (region) has a negative prejudice against the public image of entrepreneurs, the utility of entrepreneurial activities is likely to be underestimated, and people will be reluctant to participate in entrepreneurial activities [26]. Therefore, a country’s (region’s) attitude towards entrepreneurship will affect entrepreneurial activities. A culture that supports entrepreneurship makes individuals feel secure in the environment and encourages them to participate more actively in entrepreneurship [27]. At this point, individuals will also appreciate the entrepreneurial career more and believe in their ability to overcome obstacles. Therefore, other things being equal, the more positive the informal system about entrepreneurship, the stronger the influence of entrepreneurial self-efficacy on entrepreneurial intention.

The importance of institutions to the entrepreneurial intention of college students is self-evident. In addition, social support perception is a force or factor that promotes human development in the social environment and is defined as the subjective feeling and evaluation of entrepreneurial individuals for their degree of support from the outside world [28]. Therefore, perception of social support is also one of the important factors affecting entrepreneurship. Existing studies have also shown that college students with a high perception of social support are more likely to engage in entrepreneurial activities. The essence of social support is the intimate relationships between people. These supports come from both tangible material support, action support, information support, and feedback support, as well as intangible spiritual support and emotional support. At the same time, social support is not only a type of one-way care or help, but also a form of social exchange and social interaction between people in most cases [29].

Based on the above research background, this paper takes the university town of Jinan City, Shandong Province as a case area, and systematically studies the relationship between college students’ non-cognitive ability and entrepreneurial intention from a multi-dimensional perspective, as well as the mediating role of social support perception in the relationship between the two, with a view to fundamentally understand the factors and mechanisms that affect college students’ entrepreneurial willingness and provide entrepreneurial guidance and services to college students in a targeted manner. At the same time, it provides decision-making reference for promoting social harmony, stability, economic health, and sustainable development.

## 2. Theoretical Framework

### 2.1. Concept Definition

#### 2.1.1. The Definition of Non-Cognitive Ability

In psychology, non-cognitive ability is considered a personality trait, and refers to the psychological factors that are displayed by workers; they have an important impact on individual social, economic, life, and other behaviors. It is widely used in psychology, economics, education, and other fields [29,30]. Based on the Big Five Personality Scale, this study divided non-cognitive abilities into five dimensions: openness, conscientiousness, extraversion, agreeableness, and emotional stability [31,32]. Openness refers to the tendency to accept new things; the more open individuals are, the stronger their innovation ability and curiosity [31,32]. Conscientiousness refers to having a sense of responsibility and diligence; the more conscientious the individual is, the stronger the persistence of their goal-oriented behavior, and the higher their sense of achievement in their learning or career [31,32]. Extraversion means that the individual’s attention is not focused on the subjective inner world, but on the external world of people and things; the stronger the extroversion, the more positive their attitude towards facing challenges and the stronger their social ability [31,32]. Agreeableness refers to an individual’s tendency to deal with affairs in a cooperative and selfless manner; the stronger the agreeableness, the easier it is to resonate, empathize, and gain more trust and support in interpersonal communication [31,32]. Emotional stability refers to the predictability and consistency of emotional feedback, and individuals with strong stability experience no drastic emotional changes [31,32].

#### 2.1.2. The Definition of Entrepreneurial Intention

Entrepreneurial intention is the subjective attitude of possible entrepreneurs regarding whether to carry out entrepreneurial activities, and it is a good predictor of entrepreneurial behavior [33]. This study measures entrepreneurial intention from the three dimensions of entrepreneurial feasibility, entrepreneurial propensity, and entrepreneurial desirability [34]. Entrepreneurial feasibility includes personal control and a sense of responsibility. Personal control is a self-influencing process in which individuals achieve their expected goals through self-guidance and self-motivation, and this process mainly focuses on individual behavioral norms [35]. Responsibility awareness is the psychological characteristic of individuals consciously and conscientiously fulfilling their responsibilities in the process of starting a business and transforming responsibility into actions. This is a type of conscious awareness and an indispensable personal ability in the entrepreneurial process. Entrepreneurial propensity refers to the probability that entrepreneurial behavior may occur—that is, an individual’s subjective willingness to engage in entrepreneurial activities in the future [36]. The entrepreneurial behavior tendency is the key link between entrepreneurial intention and formal entrepreneurial behavior, which can provide a more reliable basis for predicting whether an individual can truly start a business in the future [37]. Entrepreneurial desirability includes innovation orientation and achievement orientation. The first trait is innovation orientation. Entrepreneurship is the process of realizing innovation. The premise for entrepreneurs to obtain profits is to use new business models, new technologies, new services, and new products to meet market demands, thereby creating unique value. The second trait is achievement orientation, which shows that individuals pay attention to the consequences, efficiency, and standards, pursue the improvement of products or services, and strive to optimize the use of resources in the organization. Achievement orientation is also an important part of entrepreneurship [38].

#### 2.1.3. Definition of Social Support Perception

Social support perception is an individual’s sense of support and care from others [28,39]. For college students and college graduates, the perception of social support specifically involves the material and spiritual support received from five types of people: family, relatives, friends, classmates, and lovers [40]. This paper selects the Perceived Social Support Scale (PSSS) compiled by Zimet, which is used to measure the degree of individual perception of support from various social support sources, and the total score reflects the total degree of social support felt by the individual [41]. In other words, the perception of social support is measured from the perspective of three sources of social support, namely family, friends, and others, emphasizing the individual’s understanding and comprehension of various sources of social support. Family support refers to the material and spiritual support that parents and relatives can provide to college students, such as providing entrepreneurial capital support, assisting in entrepreneurial decisions, and spiritual incentives. Friends support is mainly the encouragement and help given by friends when entrepreneurs encounter difficulties, including emotional support, information support, instrumental support, etc. Support from others can be understood as the help that individuals receive from teachers, relatives, classmates, colleagues, and other important individuals through social connections to reduce psychological stress, relieve mental tension, and improve adaptability.

### 2.2. Theoretical Analysis

#### 2.2.1. The Direct Influence of Various Dimensions of Non-Cognitive Ability on Entrepreneurial Intention

As a type of implicit human capital, non-cognitive ability plays a significant role in promoting individuals’ entrepreneurial intention and entrepreneurial decision-making [42]. Openness is a concentrated expression of individual wisdom and creativity. The entrepreneurial ability of individuals is closely related to their innovation ability, and individuals with a high sense of achievement, self-confidence, creativity, pressure resistance, independent preference, and other non-cognitive factors are more likely to accept new things. Therefore, such entrepreneurs with open endowment tend to possess more innovative spirit and stronger entrepreneurial intention [43,44,45]. Conscientiousness is the embodiment of responsibility and perseverance. Due to the coexistence of high returns and high risks in entrepreneurial activities, only individuals with strong conscientiousness face risks such as capital loss and entrepreneurial failure, and higher entrepreneurial returns can help to stimulate their personal entrepreneurial intention [43,44,45]. Extroversion includes the characteristics of being active and lively, possessing strong social skills, and being helpful and hopeful regarding the future, and these characteristics are helpful to enhance the confidence and hope of entrepreneurial success and enhance personal entrepreneurial motivation and willingness [43,44,45]. Agreeableness is reflected in being able to gain the trust and support of others. For entrepreneurial activities, entrepreneurs need to be more decisive and determined to make decisions. However, the rapidly changing social and market environment has put forward increasingly high requirements for the entrepreneur’s decision-making and execution ability. In this process, the more the entrepreneur can gain the trust of others, the more able they are to make and execute decisions [43,44,45]. Emotional stability is mainly manifested in the absence of obvious emotional fluctuations. Entrepreneurs with stable emotions are not easily affected by negative emotions, nor are they prone to anxiety and depression, and their entrepreneurial intentions are easy to achieve [43,44,45]. In conclusion, hypothesis H1 is proposed: all dimensions of non-cognitive ability have a positive effect on entrepreneurial intention.

#### 2.2.2. The Mediating Effect of Social Support Perception on Non-Cognitive Ability Dimensions and Entrepreneurial Intention

1.The direct influence of non-cognitive ability on social support perception

Each dimension of non-cognitive ability is closely related to social support perception [46]. Openness tends to be conducive to innovation rather than being constrained by the current environmental resources; individuals will identify and make use of various opportunities, and it is easier to perceive social support in a timely manner [47,48]. In terms of conscientiousness, the emergence of conscientious behavior can bring more pleasure to others, and the pleasure of others can also lead individuals to actively obtain more social support [46]. In terms of extraversion, entrepreneurs with strong extraversion expand their social capital by building interpersonal networks and improving their social communication ability. They usually have relatively strong interpersonal communication ability and resource transformation ability, and the interpersonal network circle is often more diversified, so it can provide increasingly higher-quality external support for entrepreneurial activities [46,48]. In the aspect of humanity, the more cooperative entrepreneurs are, the more honestly they can communicate with people, and their ability to obtain and maintain social support and control their emotions and individual feelings is usually outstanding [46,48]. In terms of emotional stability, entrepreneurs who can easily understand and accept others and whose emotions are not affected by the outside world are less likely to deny themselves, and their social support perception is also stronger [46,48]. Based on the above analysis, the following hypothesis is proposed: H2: non-cognitive abilities have a positive effect on the perception of social support.

2.The direct impact of social support perception on entrepreneurial intention

Entrepreneurship is a social activity, and choosing to start a business is a major decision in the career planning of college students. Therefore, college student entrepreneurs will seek advice and support from those around them [29,49]. When college students face a complex entrepreneurial environment, good social support is also conducive to better reducing entrepreneurial pressure, adapting to the entrepreneurial environment, and increasing their entrepreneurial rate [50]. For entrepreneurial feasibility, the perception of social support is helpful for college students to adjust psychologically, enhance their self-efficacy, strengthen their personal control, and improve their ability to deal with negative events [36,49]. For entrepreneurial tendency, the perception of social support, as an important factor affecting people’s physical and mental health, can enhance the ability of individuals to respond to stressful situations [36,49]. For entrepreneurial aspiration, positive opinions of others can allow entrepreneurs to have a higher evaluation of their ability to control and cope with multiple tasks [37,49]. Therefore, the support environment in which the individual lives and the level of support that the individual perceives will directly affect the individual’s corresponding changes in their own behavior and decision-making. Based on the above analysis, the following hypothesis is proposed: H3: the perception of social support has a positive effect on entrepreneurial intention.

3.Mediating effect between perception of social support and the influence of non-cognitive ability on entrepreneurial intention

The degree to which individuals perceive that they are being supported and cared for by others is not only directly related to the quantity or quality of their supportive interactions with others, but is also affected by various dimensions of the non-cognitive abilities of the recipients [46]. Social support perception can also guide individuals to reduce their psychological stress, reduce tension, and enhance their adaptability [28], so as to stimulate greater entrepreneurial enthusiasm and entrepreneurial intention among college students [46]. When an individual’s strong non-cognitive ability is coupled with entrepreneurial support from important groups (parents, teachers, friends, etc.), their entrepreneurial intention will be significantly improved [51]. In other words, while non-cognitive ability directly affects entrepreneurial intention, it also indirectly affects entrepreneurial intention through social support perception [52]. Therefore, the three variables have a logical relationship of “non-cognitive ability (openness, conscientiousness, extraversion, agreeableness, and emotional stability) → social support perception → entrepreneurial intention”. Accordingly, the following research hypothesis is proposed: H4: the perception of social support plays a mediating role between non-cognitive ability and entrepreneurial intention.

To summarize, this paper attempts to incorporate non-cognitive ability, social support perception, and entrepreneurial intention into the same analysis framework (Figure 1). It is intended to identify the functional path of non-cognitive ability and social support perception with regard to entrepreneurial intention by constructing the structural relationship model of “non-cognitive ability—social support perception—entrepreneurial intention” in relation to college students. The model consists of two parts: the first part is the mechanism of non-cognitive ability’s effect on entrepreneurial intention; the second part is the mediating effect of social support perception between non-cognitive ability and entrepreneurial intention (Figure 1).

## 3. Method

### 3.1. Participants

The empirical research data in this paper were obtained from questionnaires. The “Questionnaire for entrepreneurial intention of College Students in Jinan” was created, which includes a total of 77 items of investigation. As the provincial capital city, Jinan has a certain representativeness of the whole country, so the research conclusions obtained by taking Jinan as a case area have certain reference value for the exploration of the entrepreneurship of college students in large- and medium-sized cities throughout the country.

This study adopted a combination of stratified sampling and random sampling for research. First, we took districts as the primary sampling unit in Jinan City; according to the regional economic development status, four districts—Lixia District, Zhangqiu District, Licheng District, and Changqing District—were selected as the sample survey areas. Among them, Lixia District had the highest economic development level in Jinan City, Zhangqiu District and Licheng District were in the middle level, and Changqing District had a relatively weak economic development level. Second, one undergraduate and one college were randomly selected in each district. Finally, 40–70 college students were randomly selected from each sample college according to a certain proportion, and the survey was conducted in the form of one-on-one interviews. Data collection was divided into two stages: pre-investigation and formal investigation. From July to September 2021, the pre-investigation stage was carried out. One hundred questionnaires were collected. After reliability and validity analysis, some items were improved. In order to ensure the effectiveness of the questionnaire, the research team orally modified the questionnaire questions. During October–December 2021, the formal research stage took place, wherein 600 questionnaires were officially submitted, and 450 questionnaires were considered valid. The sample descriptive statistics are shown in Table 1.

### 3.2. Instruments

Based on domestic and foreign mature scales, combined with the research objectives, the scales of non-cognitive ability, social support perception, and entrepreneurial intention were designed. Each variable used a 5-point Likert scale, with “1” to “5” representing “strongly disagree”, “disagree”, “generally”, “agree” and “strongly agree”, respectively.

#### 3.2.1. Independent Variable: Non-Cognitive Ability

According to the “Big Five personality” model, with reference to the relevant research of Liu Chuanjiang [53], Roger [54], and Neneh [55], and in combination with the Revised NEO (Neuroticism-Extraversion-Openness) Personality Questionnaire, the non-cognitive ability questionnaire was designed, focusing on the five dimensions of openness, conscientiousness, extraversion, agreeableness, and emotional stability (Table 2).

#### 3.2.2. Mediating Variable: Perception of Social Support

We selected the Perceived Social Support Scale (PSSS) compiled by Zimet. The Perceived Social Support Scale is divided into three dimensions and twelve items, including family support, friends’ support, and others’ support. It is used to measure the individual’s perception of the support received from family members, friends, and other people, and to reflect the total social support felt by the individual with the total score. It is shown in Table 3.

#### 3.2.3. Dependent Variable: College Students’ Entrepreneurial Intention

Referring to the entrepreneurial problems of college students discussed by Kim and Park, and the entrepreneurial intention scale developed by Fan Wei and Wang Chongming, in this paper, entrepreneurial intention was evaluated using 25 questions focusing on three dimensions, namely entrepreneurial feasibility, entrepreneurial propensity, and entrepreneurial desirability [56,57,58], as shown in Table 4.

### 3.3. Procedure

All study procedures were approved by the researchers’ institutional review board and the school’s administration. On the day of college student data collection, trained study staff (graduate researchers and undergraduate research assistants) conducted random interviews on campus. All 600 (100%) college students provided consent and completed the survey. Survey items were read aloud by researchers while college students responded on their own paper copy. Other members of the research team were available to answer questions and provide assistance as needed. Surveys were typically completed within 30 min. For college student participation, each student received a $1 donation for school supplies.

### 3.4. Data Analysis

This study applied Amos21.0 (IBM, New York, NY, USA) to test and analyze the internal mechanism of the impact of non-cognitive ability and social support perception on entrepreneurial intention. The structural equation model consists of three latent variables, namely non-cognitive ability, social support perception, and entrepreneurial intention, each of which was measured by multiple items in the scale. In order to further verify the mediating effect and influence mechanism of social support perception, the bootstrap method was used.

## 4. Empirical Test and Result Analysis

### 4.1. Analysis of Group Differences

#### 4.1.1. Analysis of Variance

One-way analysis of variance (one-way ANOVA) was used to compare the means of multiple samples in a completely random design. Its statistical purpose is to infer whether the means of the population represented by each sample are equal. If the *p*-value is less than 0.05, it indicates that there is a significant difference.

1.Differences in entrepreneurial intention of different grades

Through one-way analysis of variance, we compared the entrepreneurial intention of different grades and obtained the above results. There were significant differences in entrepreneurial intention among different grades (*p* < 0.05). From the average value, it can be seen that the entrepreneurial intention of senior students is stronger (Table 5).

2.Differences in entrepreneurial intention with frequent participation in entrepreneurial forum lectures

Through one-way analysis of variance, the above results were obtained by comparing the willingness to participate in entrepreneurial forum lectures. There was a significant difference in the willingness to start a business depending on whether the respondents often participated in the entrepreneurial forum lectures (*p* < 0.05). From the average value, it can be seen that those who regularly participate and those who always participate have stronger entrepreneurial intention (Table 6).

#### 4.1.2. *T*-test

1.Analysis of differences in entrepreneurial intention in different school types

Using a *t*-test to compare the entrepreneurial intention in different school types, the above results were obtained. There were significant differences in entrepreneurial intention among different school types (*p* < 0.05). It can be seen from the average that the number of college students was larger than that of undergraduate students (Table 7).

2.Analysis of differences in entrepreneurial intention at the university level and above regarding participation in college student entrepreneurial competitions and winning awards

Through the *t*-test, the entrepreneurial intention in terms of participation in college entrepreneurship competitions and winning awards at the school level and above was compared, and the above results were obtained (*p* < 0.05). From the average value, it can be seen that those who have participated in an entrepreneurship competition are more willing to start a business than those who have not (Table 8).

### 4.2. Reliability and Validity Test

#### 4.2.1. Reliability Test

After reliability analysis of 48 items for the 5 dimensions of the independent variable of non-cognitive ability, 12 items of the intermediary variable of social support perception, and 17 items of the dependent variable of entrepreneurial intention, the Cronbach α coefficients were found to all be greater than 0.7, indicating that this part of the questionnaire had good reliability (Table 9).

#### 4.2.2. Validity Test

As shown in Table 10, the KMO (Kaiser-Meyer-Olkin) values of non-cognitive ability, perception of social support, and entrepreneurial intention were 0.961, 0.921, and 0.917, which are all greater than 0.70, indicating that the questionnaire was suitable for factor analysis. The Bartlett sphericity test results showed that the significant probability corresponding to the approximate chi-square value was 0.000 (*p* < 0.01), so the validity structure was good. The total variance explained rate was 68.717% and 68.16%, greater than 60%, so the validity of the scale was considered to be good. The loading of each measurement item was higher than 0.5, there was no double factor loading, and the measurement items under each dimension were aggregated according to the theoretical distribution, indicating that the questionnaire had good content validity (Table 10).

As shown in Table 11, confirmatory factor analysis results showed that the standardized factor loading of each item was greater than 0.5, and the standard error value S.E. was also less than the standard of 0.5, which proved that the validity of the questionnaire was good. At the same time, the AVE (Average Variance Extracted) of each dimension was greater than 0.5, and the square root of the AVE was greater than the correlation coefficient between the variables, indicating that the scale had good convergent and discriminant validity among the variables (Table 11).

### 4.3. Correlation Analysis

In order to verify the interaction between multiple variables, it is necessary to carry out correlation analysis. If the correlation coefficient is positive and passes the significance test, there is a significant positive correlation between the variables; if the correlation coefficient is negative and passes the significance test, there is a significant negative correlation between the variables. This study conducted a pairwise correlation analysis on the dimensions of non-cognitive ability, social support perception, and entrepreneurial intention. The results are as follows (Table 12).

The above table shows the results of the correlation analysis. The *p*-values corresponding to the correlation coefficients between each dimension of non-cognitive ability, perception of social support, and entrepreneurial intention were all less than 0.05, which is statistically significant, indicating that there are significant correlations between the studied dimensions, namely, the perception of social support and entrepreneurial intention. Subsequent impact relationship analysis was performed.

### 4.4. Structural Equation Model Fitting Index

The fitting index of the structural equation model showed that the value of X^2^/df was 1.230, which is less than 3. The RMSEA (Root Mean Square Error of Approximation) was 0.023, which is less than the standard level of 0.08, indicating a good fit. GFI (Goodness of Fit Index) = 0.884, AGFI (Adjusted Goodness of Fit Index) = 0.873, NFI (Normed Fit Index) = 0.904, IFI (Incremental Fit Index) = 0.980, CFI (Comparative Fit Index) = 0.980, TLI (Tucker-Lewis Index) = 0.979. All goodness-of-fit indicators met the general standard, indicating that the structural equation model established in this study was effective and consistent with the recovered data. The match was better.

### 4.5. Analysis of Data Results

Through the path analysis of the structural equation model, the path coefficient value and C.R. value of the structural equation model were obtained. The results are shown in Table 13.

#### 4.5.1. Path Analysis of the Impact of Various Dimensions of Non-Cognitive Ability on Entrepreneurial Intention

The standardized path coefficients of openness, conscientiousness, extraversion, and emotional stability to entrepreneurial intention were 0.176, 0.185, 0.28, and 0.216, respectively, which were significantly established at the level of 0.001, indicating that these factors have significant positive effects on entrepreneurial intention. Among them, extraversion has the greatest impact on entrepreneurial intention.

The standardized path coefficient of agreeableness on entrepreneurial intention was 0.018 (*t* value = 0.266, *p* = 0.79 > 0.05), indicating that agreeableness has no significant effect on entrepreneurial intention. Theoretically speaking, college students with a more agreeable personality can obtain more understanding and support, so that they can start a business more smoothly. However, from the current data, this finding may be related to the fact that the local environment causes college students to feel unfairly treated and dissatisfied, and they do not have a sense of belonging to, identity with, or dependence on the local environment, thus affecting their willingness to start a business there. College students with a high level of agreeableness have an optimistic and positive attitude towards human nature, and they believe that the mentality of human nature will promote their subjectively more supported positive emotional experience, but they may lack entrepreneurial motivation.

#### 4.5.2. Analysis of the Impact Path of Each Dimension of Non-Cognitive Ability on Perception of Social Support

The standardized path coefficients of openness, conscientiousness, extroversion, agreeableness, and emotional stability on social support perception were 0.142, 0.148, 0.17, 0.214, and 0.212, respectively, indicating that these factors have significant positive effects on social support perception. Extraversion, agreeableness, and emotional stability are significant at the level of 0.001, openness is significant at the level of 0.01, and conscientiousness is significant at the level of 0.05.

#### 4.5.3. Path Analysis of the Impact of Social Support Perception on Entrepreneurial Intention

The standardized path coefficient of social support perception on entrepreneurial intention is 0.446, indicating that social support perception has a significant positive effect on entrepreneurial intention, which is significantly established at the level of 0.001.

#### 4.5.4. Analysis of the Mediating Effect of Social Support Perception

This study used the bootstrap method to test the mediating effect of social support perception among the five dimensions of non-cognitive ability and entrepreneurial intention. The analysis results of the intermediary effect are shown in Table 14.

1.The mediating effect of social support perception on the effect of openness on entrepreneurial intention

The effect value of openness on entrepreneurial intention through social support perception is 0.063, the 95% confidence interval is [0.018–0.12], and the *p*-value is less than the significant level of 0.05, indicating that there is an intermediary effect, so the hypothesis is tenable. As the independent variable is significant for the dependent variable, it is still significant after adding the intermediary variable, which confirms that social support perception plays a part in the mediating role of openness in entrepreneurial intention.

2.The mediating effect of social support perception on the effect of conscientiousness on entrepreneurial intention

The effect value of due diligence on entrepreneurial intention through social support perception is 0.066, the 95% confidence interval is [0.012–0.129], excluding 0, and the *p*-value is less than the significant level of 0.05, indicating that there is an intermediary effect, so the hypothesis is tenable. As the independent variable is significant for the dependent variable, it is still significant after adding the mediator variable, indicating that it is a partial mediator.

3.The mediating effect of social support perception between extraversion and entrepreneurial intention

The effect value of extraversion influencing entrepreneurial intention through social support perception is 0.076, the 95% confidence interval is [0.032–0.145], excluding 0, and the *p*-value is less than the significant level of 0.05, indicating that there is an intermediary effect, so the hypothesis is tenable. As the independent variable is significant for the dependent variable, it is still significant after adding the intermediary variable, which indicates that it is part of the intermediary.

4.The mediating effect of social support perception between agreeableness and entrepreneurial intention

The effect value of agreeableness affecting entrepreneurial intention through perception of social support is 0.095, the 95% confidence interval is [0.037–0.176], excluding 0, and the *p*-value is less than the significant level of 0.05, indicating that there is a mediating effect, so the hypothesis is established. As the independent variable is not significant for the dependent variable, the mediation is significant after adding the mediator variable, indicating that it demonstrates a complete mediation.

5.The mediating effect of social support perception on the influence of emotional stability on entrepreneurial intention

The effect value of emotional stability affecting entrepreneurial intention through the perception of social support is 0.094, the 95% confidence interval is [0.047–0.171], excluding 0, and the *p*-value is less than the significant level of 0.05, indicating the existence of a mediating effect, so the hypothesis is established. As the independent variable is significant for the dependent variable, it is still significant after adding the mediator variable, indicating that it is a partial mediator.

## 5. Discussion

This paper focuses on the problem of college students’ entrepreneurial intention and constructs a theoretical analysis framework of “non-cognitive ability—social support perception—entrepreneurial intention”. Based on field research and using a structural equation model, we analyzed the impact of various dimensions of non-cognitive ability (openness, conscientiousness, extroversion, agreeableness, and emotional stability) on entrepreneurial intention. On this basis, the bootstrap method was used to further verify the mediating effect of social support perception on the influence of various dimensions of non-cognitive ability on entrepreneurial intention.

### 5.1. Openness, Conscientiousness, Extroversion, and Emotional Stability Have Significant Positive Effects on Entrepreneurial Intention

The mechanisms of action may be as follows. More open entrepreneurs are willing to accept new things and tend to invest time and energy in identifying entrepreneurial opportunities, analyzing entrepreneurial risks, finding entrepreneurial partners, and formulating entrepreneurial plans. However, those who are less open may have scattered energy, weak motivation, and a tendency to retreat from difficulties encountered in achieving the goal of entrepreneurship. Individuals with a high sense of responsibility have a strong sense of responsibility for entrepreneurial work, work hard, do things efficiently, do things in an orderly manner, and strive to achieve their entrepreneurial goals. The more conscientious college students are, the more willing they are to start a business, because they have the elements of entrepreneurship. Extroverted college students are willing to participate in interpersonal communication, have the courage to undertake more social activities, and can better integrate into local social life. This will also improve the individual’s ability to bear entrepreneurial risks and reduce loss aversion, thus contributing to the growth of individual entrepreneurial intention. College students with strong emotional stability have strong self-regulation ability and have the advantage of eliminating negative emotions regarding entrepreneurship in time, so as to ensure the full exploitation of their personal entrepreneurial ability and stabilize their entrepreneurial intention.

### 5.2. All Five Dimensions Have a Significant Positive Impact on Perception of Social Support

Their impact paths may be as follows. Extroverted college students tend to have stronger interpersonal skills, and their social support sources are wider and higher in quality, so the perception of social support obtained is also more significant. College students with stronger agreeableness have a stronger sense of cooperation and responsibility, a stronger sense of support from others, and a stronger sense of social support. College students who are emotionally stable tend to have a calm personality, can calmly deal with problems that they encounter and seek help, and can detect the support and help provided by others in a timely manner. College students with strong openness are more creative and have the ability to recognize and utilize various opportunities, and they can sense the emergence of social support and make use of it in time. The sense of responsibility and the conscientious behavior of entrepreneurs can bring a more positive emotional experience to the relevant groups, so as to obtain positive social feedback and allow them to experience more social support.

### 5.3. Perception of Social Support Has a Positive Impact on Entrepreneurial Intention

First, the stronger the social support obtained, the more the college student entrepreneurs can affirm themselves and strengthen their sense of responsibility under the support of important others, so as to adhere to their original entrepreneurial intention and maintain a relatively strong entrepreneurial intention. Second, when college students deal with crisis and stress events, social support perception enables them to better adapt to their environment, relieve pressure, and reduce the withdrawal psychology of entrepreneurs. Third, active social support can enhance the innovation level and achievement level of college student entrepreneurs. At the same time, as entrepreneurs, when college students are concerned about respecting other people’s feelings and needs, they will also more firmly respect their own feelings and needs, thereby affirming the individual’s entrepreneurial intention and promoting entrepreneurial activities.

### 5.4. Social Support Perception Acts as a Mediator in the Influence of Non-Cognitive Ability on Entrepreneurial Intention

The perception of social support in the influence of openness, conscientiousness, extroversion, and emotional stability on entrepreneurial intention is partially mediated; in the influence of agreeableness on entrepreneurial intention, it shows a complete mediating role. Its mechanisms of action may be as follows. Openness indirectly affects entrepreneurial intention by influencing social support perception to a certain extent. Specifically, college students with higher openness tend to have a stronger social support perception, and a relatively strong social support perception can stimulate college students’ entrepreneurial intention. College students with high conscientiousness can influence their entrepreneurial intention by influencing their social support perception. These students can better achieve the emotional experience of being understood, respected, and supported in society because of their degree of self-control, completing the generation of behaviors that affect the entrepreneurial tendency, and they have a positive attitude towards entrepreneurial intention promotion. The college students with a high extraversion level are perceived to be psychologically strong due to their extraverted personality characteristics, enthusiasm, optimism, vitality, and positive emotions. It is also this psychological strength that means that they have more positive emotions that are supported and understood subjectively, so as to increase their interaction with people, thereby improving their entrepreneurial intention and promoting the generation of entrepreneurial behavior. College students with a high level of agreeableness maintain a positive and optimistic attitude towards people, have full trust in their interpersonal relationships, and believe that human nature is inherently good. Such people are more likely to obtain, trust, and use support. These supports are transformed into entrepreneurial behavior tendencies and the desire for entrepreneurial behavior improves the conversion rate of entrepreneurial intention to a certain extent. College students who are emotionally stable have a large number of positive experiences, and most of them have a positive attitude towards objective factors; emotionally stable personalities are not easily affected by the external environment, individuals can remain alert and rational, and they tend to have higher social status. Support perception, and relatively deeper social support perception, stimulate greater entrepreneurial intention.

Finally, due to the limitation of the research group, the results of this research must also have certain limitations, which are expected to be improved in future research. In terms of research samples, only universities in Jinan City, Shandong Province, were selected for research. How to further improve the research framework, increase the number of spatial samples, and the number of questionnaires remains to be further explored. Future research can expand the sample size or conduct comparative studies on different types of samples in different regions to increase the applicability of the research conclusions. In the follow-up consideration, the entrepreneurial willingness will be divided into multiple dimensions, and the internal mechanism that affects the entrepreneurial willingness will be discussed in depth.

## 6. Conclusions

This study verifies that both non-cognitive ability and the perception of social support have an impact on entrepreneurial intention, and they are involved in a sequential development process. College students’ perceptions of social support are affected by individual non-cognitive abilities, which will continue to affect their entrepreneurial intention. Therefore, there are two paths by which college students’ non-cognitive ability can influence their entrepreneurial intention. First, the four dimensions of non-cognitive ability, namely openness, conscientiousness, extroversion, and emotional stability, can directly affect entrepreneurial intention. Second, the five dimensions of non-cognitive ability, namely openness, conscientiousness, extraversion, agreeableness, and emotional stability, affect entrepreneurial intention through the mediating role of perceived social support. According to the research conclusions, the following suggestions are put forward according to three levels: college students, colleges and universities, and the government.

### 6.1. College Students Should Cultivate the Characteristics of Innovation and Entrepreneurship and Improve Their Entrepreneurial Ability

From the results of this research, college students’ high levels of openness, conscientiousness, extroversion, and emotional stability have a significant positive impact on entrepreneurial intention. The entrepreneurial process is complex and volatile, and there are many uncertainties. Therefore, entrepreneurs are required to have personal characteristics such as good literacy, sufficient entrepreneurial skills, and tempered entrepreneurial practices. In other words, college students must have the qualities of perseverance, courage to make breakthroughs, and resilience towards hardships, as well as the strength to maintain self-confidence and resist setbacks. Therefore, college students must purposefully cultivate these qualities in their studies and life, carry out comprehensive and systematic exercises, and build a reliable foundation for entrepreneurship. To summarize, college students should fully exploit their own advantages, learn comprehensively with multiple resources, and participate in practical activities independently to cultivate their entrepreneurial characteristics and optimize their entrepreneurial skills.

### 6.2. Colleges and Universities Should Improve the Quality of Entrepreneurship Education and Promote Entrepreneurial Actions

Institutions of higher learning should attach great importance to entrepreneurship and innovation education, and they should focus on cultivating innovative and highly skilled individuals who keep pace with the times. We must comprehensively revise the training program for innovative talent; integrate innovation and entrepreneurship education into professional education, ideological and political education, and other education; and build a curriculum system that pays equal attention to “professional ability + project orientation + innovation and entrepreneurship module”. We should promote teaching research and education reform by rewarding innovation and entrepreneurship, and we should encourage teachers to take the initiative to meet the needs of enterprises and carry out innovation and entrepreneurship activities, so as to create an innovation and entrepreneurship teaching team with “school–enterprise interoperability, combination of professional and part-time”. We may set up an off-campus entrepreneurship tutor library to include outstanding alumni and entrepreneurs. We should strengthen school enterprise cooperation and establish a sound entrepreneurial practice system. It is far from sufficient to rely solely on the faculties of colleges and universities. Schools should invite company experts to impart some experience to students and address their doubts. They should also lead students to the company to observe and participate in internships, so that students can effectively improve their ability to solve practical problems in the established training base and enhance the practical effect of innovation and entrepreneurship. It is also necessary to integrate school resources and build a cultural environment for innovation and entrepreneurship. Universities and colleges can regularly invite entrepreneurs and outstanding alumni to give lectures and make reports by holding entrepreneurship competitions inside and outside the campus, exhibit innovation achievements, and hold special lectures on entrepreneurship, alumni salons, and other activities, so as to introduce excellent cases of successful entrepreneurship into campus culture and create a cultural circle of innovation and entrepreneurship. At the same time, a variety of entrepreneurship education activities should be held, such as the Maker Culture Festival, the entrepreneurship planning competition, and the speech contest around the theme of entrepreneurship, etc. Relying on these diversified entrepreneurial cultural activities, we may achieve integration with entrepreneurial education to improve the promotion of an entrepreneurial culture. In addition, it should be noted that the target of the entrepreneurial initiative is not all college students, and it should not blindly encourage college students’ entrepreneurial willingness to avoid adverse effects. For college students with a certain entrepreneurial willingness, they can cultivate their entrepreneurial thinking, innovative ideas, and innovative entrepreneurial skills in an all-around way, thereby promoting progress in innovative and entrepreneurial work.

### 6.3. Create a Good Entrepreneurial Environment for College Students to Start a Business and Improve Their Willingness to Start a Business

Society should create a good entrepreneurial environment for college students, including institutional, financial, and cultural environments. First, it should improve entrepreneurship regulations, bankruptcy regulations, and intellectual property protection regulations; escort entrepreneurs; strengthen national supervision; and promote fair competition. A strong legal system can improve the efficiency of business transactions and reduce transaction costs, thereby enabling individuals to profit from business activities and stimulating their entrepreneurial desire. Second, in the initial stage of entrepreneurship, due to the uncertainty and high risk of entrepreneurship, financial institutions are less likely to provide financial support to entrepreneurs. Therefore, improving the financial environment and creating more diversified financing channels for college students’ entrepreneurs can improve their entrepreneurial willingness. Finally, human capital is the most active and positive factor for innovation-driven development. Therefore, national general education should be vigorously strengthened in order to provide better conditions for college students entrepreneurs to create and operate enterprises. In the context of a culture that supports entrepreneurship, individuals will feel the supportive attitude of the entire society towards entrepreneurship. The cultural environment that supports entrepreneurship makes individuals feel the sense of security brought by the environment, which will stimulate individuals’ confidence in successful entrepreneurship, improve their entrepreneurial willingness, and promote their more active participation in entrepreneurship. Therefore, a social and cultural environment that supports entrepreneurship should be created.

## Figures and Tables

**Figure 1 ijerph-19-11981-f001:**
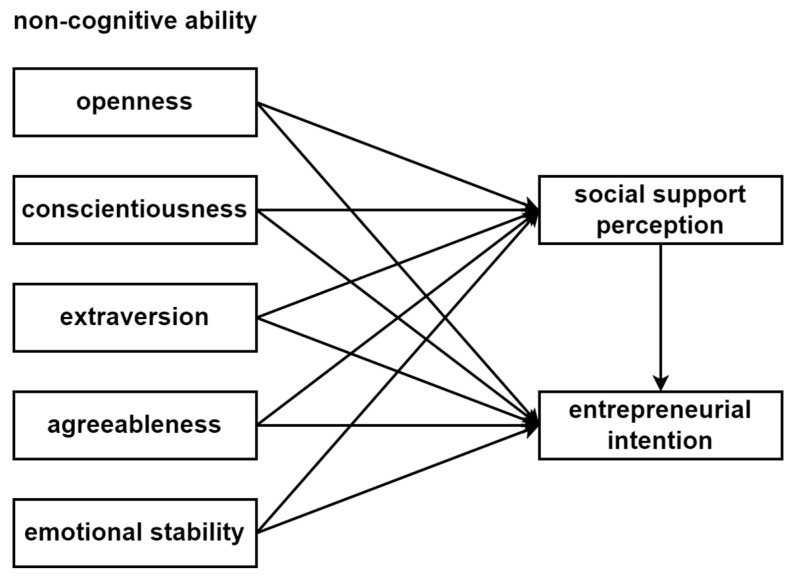
Theoretical framework diagram.

**Table 1 ijerph-19-11981-t001:** Descriptive statistics of sample.

Project	Category	Frequency	Percentage
Gender	Male	234	52
	Female	216	48
Grade	Freshman	93	20.7
	Sophomore	105	23.3
	Junior year	99	22
	Senior year	153	34
Type of School	Undergraduate	354	78.7
	Specialist	96	21.3

**Table 2 ijerph-19-11981-t002:** Evaluation indicators of non-cognitive ability.

Dimension	Subdimension	Index	Mean	Standard Deviation
Openness	Curiosity	C1: Interested in many different things	3.69	1.118
		C2: A thoughtful person	3.78	1.145
		C3: Hope to experience a new way of life	3.75	1.052
	Action force	C4: Likes to take on challenges	3.74	1.049
		C5: Activities organized by participating units	3.71	1.082
		C6: Have their own hobbies and be able to stick to them	3.73	1.055
	Imagination	C7: Can find smart ways to do things	3.68	1.073
		C8: Imaginative people	3.69	1.023
		C9: Be creative and come up with new ideas	3.7	1.085
Conscientiousness	Sense of responsibility	C10: I can concentrate on completing the work	3.88	0.929
		C11: Trustworthy	4.03	0.964
		C12: People around me praise me for being responsible	3.98	0.999
	Organized	C13: Be organized	4.05	1.087
		C14: Habit of keeping things neat and orderly	3.98	0.947
		C15: Have a plan	3.94	0.915
	Effort level	C16: At work, I try my best to do everything	4.01	0.963
		C17: Efficient, work from beginning to end	4.11	1.105
		C18: Perseverance and perseverance to get things done	3.97	0.964
		C19: Work hard to achieve your goals	4.03	0.971
		C20: People who constantly demand improvement	3.97	0.966
Extraversion	Social contact	C20: I like to make friends	3.69	1.192
		C21: Talkative	3.62	1.021
		C22: I will not reject attending gatherings with many people	3.7	1.116
	Decisive	C23: Dare to express one’s opinion	3.67	1.131
		C24: Strong and confident character	3.61	1.029
		C25: Affect others	3.69	1.105
	Vitality	C26: When I’m around, I’m usually not cold	3.67	1.088
		C27: Energetic	3.65	1.105
		C28: Passionate	3.69	1.141
Agreeableness	Altruism	C29: Willing to pay time cost for others	3.85	1.106
		C30: Make people around you feel at ease	3.79	1.089
		C31: Do my best to help others	3.71	1.01
	Compliance	C32: Obey social order	3.75	1.146
		C33: Willing to make friends with locals	3.73	1.038
		C34: Always be polite to others	3.77	1.101
	Trust	C35: If others have bad experiences, I will be very sympathetic	3.72	1.079
		C36: Think of people in the best way	3.78	1.085
		C37: Easy to get close to others	3.73	1.062
Emotional Stability	Anxiety	C38: Rarely feel anxious	3.86	1.076
		C39: Calm and good at dealing with pressure	3.78	1.008
		C40: Don’t worry too much	3.73	0.97
	Depression	C41: Be satisfied with yourself	3.9	1.122
		C42: Feel safe in life	3.79	1.077
		C43: Rarely unhappy in life	3.83	1.022
		C44: Stay positive despite setbacks	3.8	0.986
	Vulnerability	C45: Mood is not easy to swing	3.81	1.106
		C46: Rarely gets angry with others	3.78	1.133
		C47: Can control one’s emotions	3.8	1.038

**Table 3 ijerph-19-11981-t003:** Evaluation index of local adaptability.

Dimension	Index	Mean	Standard Deviation
Family Support	B1: My family can help me in a concrete way	3.69	1.118
	B2: I am able to get emotional help and support from my family when needed	3.78	1.145
	B3: I can talk to my family about my problems	3.75	1.052
	B4: My family is willing to help me make decisions	3.74	1.049
Friends Support	B5: My friends can really help me	3.88	0.929
	B6: I can count on my friends in times of trouble	4.03	0.964
	B7: My friends can share happiness and sadness with me	3.98	0.999
	B8: I can discuss my problems with my friends	4.05	1.087
Other Support	B9: Some people (teachers, relatives, classmates) will be by my side when I have a problem	3.69	1.192
	B10: I can share happiness and sadness with some people (teachers, relatives, classmates)	3.62	1.021
	B11: Some people (teachers, relatives, classmates) are a real source of comfort when I’m in trouble	3.7	1.116
	B12: There are people in my life (teachers, relatives, classmates) who care about my feelings	3.67	1.131

**Table 4 ijerph-19-11981-t004:** The evaluation index of entrepreneurial intention.

Dimension	Index	Mean	Standard Deviation
Entrepreneurial Feasibility	A1: I started a business because it gave me the opportunity to make a difference	3.69	1.118
	A2: I already have the interpersonal skills needed to start a business	3.78	1.145
	A3: I feel energized when working in a creative, passionate and dynamic environment	3.75	1.052
	A4: I have the self-confidence needed to start a business	3.74	1.049
	A5: I already have the organizational and management skills needed to start a business	3.71	1.082
	A6: I already have the ideas needed to start a business	3.73	1.055
Entrepreneurial Propensity	A7: I like to do challenging work	3.88	0.929
	A8: I think my experience can meet the needs of future work	4.03	0.964
	A9: I have now started to prepare to start a business	3.98	0.999
	A10: The current entrepreneurial environment is suitable for me to start a business	4.05	1.087
	A11: I am excited when there are unusual solutions to work problems	3.98	0.947
Entrepreneurial Desirability	A12: I have communicated my intention to start a business with my family or friends	3.69	1.192
	A13: I already have the teamwork skills needed to start a business	3.62	1.021
	A14: I’m already spending time learning about entrepreneurship	3.7	1.116
	A15: I already have the perseverance needed to start a business	3.67	1.131
	A16: I already have the financial conditions needed to start a business	3.61	1.029
	A17: I already have the learning skills needed to start a business	3.69	1.105

**Table 5 ijerph-19-11981-t005:** Differences in entrepreneurial intention of different grades.

Variable	Category	*N*	Mean	Standard Deviation	*F*	Salience
Entrepreneurial Intention	Freshman	93	3.262	0.588	4.687	0.003
	Sophomore	105	3.355	0.557		
	Junior year	99	3.356	0.578		
	Senior year	153	3.520	0.519		

**Table 6 ijerph-19-11981-t006:** Differences in entrepreneurial intention with frequent participation in entrepreneurial forum lecture.

Variable	Category	*N*	Mean	Standard Deviation	*F*	Salience
Entrepreneurial Intention	Never participate	88	3.283	0.604	3.362	0.01
	Participate occasionally	198	3.358	0.545		
	Uncertain	45	3.366	0.504		
	Participate often	63	3.493	0.602		
	Always participate	56	3.591	0.508		

**Table 7 ijerph-19-11981-t007:** Analysis of differences in entrepreneurial intention in different school types.

	Category	N	Mean	Standard Deviation	*t*	*p*
Entrepreneurial Intention	Undergraduate	354	3.3544	0.56121	−2.741	0.006
	Specialist	96	3.5306	0.5491		

**Table 8 ijerph-19-11981-t008:** Analysis of differences in entrepreneurial intention at school level and above regarding participation in college student entrepreneurship competitions and winning awards.

	Category	*N*	Mean	Standard Deviation	*t*	*p*
Entrepreneurial Intention	Yes	164	3.487	0.546	2.733	0.007
	NO	286	3.338	0.566		

**Table 9 ijerph-19-11981-t009:** Reliability analysis of variables.

Variable/Dimension	Number of Items	Cronbach’s Alpha
Entrepreneurial Intention	17	0.88
Entrepreneurial Feasibility	6	0.904
Entrepreneurial Propensity	5	0.883
Entrepreneurial Desirability	6	0.914
Social Support	12	0.921
Family Support	4	0.882
Friends Support	4	0.9
Other Support	4	0.877
Non-Cognitive Abilities	48	0.956
Openness	9	0.944
Conscientiousness	11	0.943
Extraversion	9	0.915
Agreeableness	9	0.928
Emotional Stability	10	0.948

**Table 10 ijerph-19-11981-t010:** KMO and Bartlett test.

Variable	KMO	Bartlett’s Sphericity Test		
		Approximate Chi-Square	df	Sig.
Entrepreneurial Intention	0.917	4434.751	136	0.000
Social Support	0.921	3502.37	66	0.000
Non-Cognitive Abilities	0.961	15285.919	1128	0.000
Overall Questionnaire	0.952	24583.693	2926	0.000

**Table 11 ijerph-19-11981-t011:** Factor analysis results of the overall scale.

Measurement Item	Ingredients										
	1	2	3	4	5	6	7	8	9	10	11
A1							0.778				
A2							0.791				
A3							0.8				
A4							0.745				
A5							0.786				
A6							0.775				
A7								0.703			
A8								0.747			
A9								0.745			
A10								0.759			
A11								0.751			
A12						0.757					
A13						0.78					
A14						0.755					
A15						0.784					
A16						0.801					
A17						0.802					
B1									0.714		
B2									0.75		
B3									0.753		
B4									0.773		
B5										0.741	
B6										0.686	
B7										0.748	
B8										0.748	
B9											0.708
B10											0.734
B11											0.703
B12											0.724
C1			0.795								
C2			0.79								
C3			0.773								
C4			0.769								
C5			0.764								
C6			0.778								
C7			0.817								
C8			0.785								
C9			0.81								
C10	0.736										
C11	0.723										
C12	0.716										
C13	0.719										
C14	0.696										
C15	0.729										
C16	0.734										
C17	0.755										
C18	0.748										
C19	0.727										
C20	0.743										
C21					0.772						
C22					0.757						
C23					0.706						
C24					0.723						
C25					0.682						
C26					0.761						
C27					0.731						
C28					0.731						
C29					0.792						
C30				0.74							
C31				0.714							
C32				0.661							
C33				0.707							
C34				0.717							
C35				0.709							
C36				0.682							
C37				0.727							
C38				0.783							
C39		0.776									
C40		0.782									
C41		0.757									
C42		0.748									
C43		0.758									
C44		0.732									
C45		0.748									
C46		0.774									
C47		0.779									
C48		0.771									
Eigenvalues	21.93	4.71	4.61	4.12	3.61	3.50	3.0	2.42	1.98	1.55	1.06
Variance Explained Rate	28.48%	6.11%	5.99%	5.35%	4.68%	4.54%	3.89%	3.15%	2.57%	2.01%	1.38%
Total Explanation Rate	68.16%										

**Table 12 ijerph-19-11981-t012:** Correlation analysis.

	1	2	3	4	5	6	7
Openness	1						
Conscientiousness	0.389 **	1					
Extraversion	0.226 **	0.337 **	1				
Agreeableness	0.412 **	0.527 **	0.313 **	1			
Emotional Stability	0.341 **	0.457 **	0.287 **	0.464 **	1		
Social Support Perception	0.373 **	0.438 **	0.349 **	0.466 **	0.444 **	1	
Entrepreneurial Intention	0.398 **	0.497 **	0.422 **	0.441 **	0.496 **	0.563 **	1

Note: ** Significantly correlated at the 0.01 level (two-sided).

**Table 13 ijerph-19-11981-t013:** Path analysis results.

Way			Standardized Path Coefficient	S.E.	C.R.	*p*
Social Support Perception	<---	Openness	0.142	0.043	2.738	0.006 **
Social Support Perception	<---	Conscientiousness	0.148	0.052	2.478	0.013 *
Social Support Perception	<---	Extraversion	0.17	0.041	3.362	***
Social Support Perception	<---	Agreeableness	0.214	0.055	3.493	***
Social Support Perception	<---	Emotional Stability	0.212	0.045	3.799	***
Entrepreneurial Intention	<---	Openness	0.176	0.028	2.941	0.003 **
Entrepreneurial Intention	<---	Conscientiousness	0.185	0.033	2.707	0.007 **
Entrepreneurial Intention	<---	Extraversion	0.28	0.028	4.548	***
Entrepreneurial Intention	<---	Agreeableness	0.018	0.034	0.266	0.79 *
Entrepreneurial Intention	<---	Emotional Stability	0.216	0.03	3.318	***
Entrepreneurial Intention	<---	Social Support Perception	0.446	0.048	5.24	***

Note: * means *p* < 0.05, ** means *p* < 0.01, *** means *p* < 0.001.

**Table 14 ijerph-19-11981-t014:** Test results of intermediary effect of bootstrap.

Parameter	Estimate	Lower	Upper	*p*
Openness-Social Support Perception-Entrepreneurial Intention	0.063	0.018	0.12	0.004
Conscientiousness-Social Support Perception-Entrepreneurial Intention	0.066	0.012	0.129	0.01
Extraversion-Social Support Perception-Entrepreneurial Intention	0.076	0.032	0.145	0.001
Agreeableness-Social Support Perception-Entrepreneurial Intention	0.095	0.037	0.176	0.001
Emotional Stability-Social Support Perception-Entrepreneurial Intention	0.094	0.047	0.171	0.000

## Data Availability

The data underlying the results presented in the study are all available. The data presented in this study are available on request from the corresponding author. The data are not publicly available due to privacy.

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
