# Peer review of "Research on the Influence of Non-Cognitive Ability and Social Support Perception on College Students’ Entrepreneurial Intention"

_ijerph, 2022, doi:10.3390/ijerph191911981_

Round 1

Reviewer 1 Report

The authors provide an interesting and useful insight concerning the driving factors of entrepreneurship, concentrating mainly on the relationship between non-cognitive ability and college students' entrepreneurial intention. Extending research on this particular matter of entrepreneurial initiatives is necessary and, in this respect, the authors’ contribution is undeniable.

However, the theoretical framework provided in this article misses the most important aspect: the economic theory of entrepreneurship.

In order to make this statement clearer and also to improve theoretical framework, some of the authors’ arguments were presented above along with the suggested alternative approaches and bibliography.

However, a study found that entrepreneurial behavior is driven by entrepreneurial intention, and without entrepreneurial intention, there would be no entrepreneurial behavior among college students. Therefore, to guide and encourage college students to start their own businesses, the first problem to be solved is to help college students to form their entrepreneurial intention. Entrepreneurial intention is the best predictor of entrepreneurial behavior.”

Of course, no human action can be initiated without intending to do it. But simply intending to become an entrepreneur, does not provide a certain type of entrepreneurial behavior. In this matter, clarifying the concept of entrepreneurial behavior is of utmost importance.

See:

Baumol, W.J. Entrepreneurship: Productive, Unproductive, and Distructive. J. Polit. Econ. 1990, 98, 893–921.

Sautet, F.E. The Role of Institutions in Entrepreneurship: Implications for Development Policy. In Mercatus Policy Series; Policy Primer no. 1; Mercatus Center: Arlington, VA, USA, 2005; pp. 1–18.

” In other words, the entrepreneurial intention of college students is inevitably affected by the external social environment, among which the perception of social support is one of the most important factors, which refers to the subjective feeling and evaluation of the degree of support from the outside. Existing studies have also shown that college students with a high perception of social support are more likely to engage in entrepreneurial activities.”

Entrepreneurship is an economic activity, it means human action initiated in an economic environment based on voluntary exchange. Social support is important, but mainly as a validator of voluntary exchange between the initiator and the potential buyer of entrepreneurial products. One can agree that social support won’t compensate indefinitely the lack of entrepreneurial success. In this aspect an extension of the external social environment is needed (and inevitable as well): 1) social environment provides the potential entrepreneurial initiatives due to individuals needs that entrepreneurs can/must anticipate as a prerequisite of entrepreneurial intentions. This argument is consistent with the view on entrepreneurship as a market process; 2) informal institutions, such as culture, can play an important part in the process of entrepreneurial intentions and behaviors.

See:

Hébert, R.F.; Link, A.N. In Search of the Meaning of Entrepreneurship. Small Bus. Econ. 1989, 39–49.

Rothbard, M N 1985, ‘Professor Hébert on Entrepreneurship,’ The Journal of Libertarian Studies, Vol. VII, No. 2, pp. 281-286.

Rothbard, M N 1987, ‘Breaking Out of the Walrasian Box: The Case of Schumpeter and Hansen,’ The Review of Austrian Economics, Vol. 1, pp. 97-108.

” Based on the above research background, this research will systematically analyze the internal and external factors that affect college students' entrepreneurial intention and the interaction between internal and external factors, so as to fundamentally understand the factors and mechanisms that affect college students' entrepreneurial intention and target them towards college students. Among them, the internal cause is the personality traits of college students—that is, non-cognitive ability—and the external cause is the social support perception of college students.”

Analyzing external factors concerning entrepreneurial intentions means considering the formal and informal institutions or, more specifically, the institutional arrangements. This study ignores completely the significance of institutions as external factors of undertaking entrepreneurial initiatives. Entrepreneurial intentions do not develop based only on personality traits or social support. It needs a proper institutional framework, as emphasized by a consistent and extended economic literature.

See:

North, D.C. Institutions. J. Econ. Perspect. 1991, 5, 97–112.

North, D.C. Understanding the Process of Economic Change; Princeton University Press: Princeton, NJ, USA, 2005; pp. 103–115. ISBN 0-691-11805-1.

Williamson, O.E. The New Institutional Economics: Taking Stock, Looking Ahead. J. Econ. Lit. 2000, 38, 595–613.

Boettke, P.J.; Coyne, C.J.; Leeson, P.T. Institutional Stickiness and the New Development Economics. Am. J.Econ. Sociol. 2008, 67, 331–358.

Sobel, R.S. Testing Baumol: Institutional quality and the productivity of entrepreneurship. J. Bus. Ventur. 2008, 23, 641–655.

Boettke, P.J.; Fink, A. Institutions first. J. Econ. 2011, 7, 499–504.

Acemoglu, D.; Johnson, S.; Robinson, J. Institutions as a fundamental cause of long-run growth. In Handbook of Economic Growth; Aghion, P., Durlauf, S.N., Eds.; North Holland Publishing Co.: Amsterdam, The Netherlands, 2005; Volume 1A, pp. 386–464. ISBN 9780444520418.

Colleges and universities should improve the quality of entrepreneurship education and promote entrepreneurial actions”

Theories of human capital are stressing on increasing the level of education as a prerequisite for development, in general or in particular matters (e.g., entrepreneurial initiatives). But as some studies emphasize, in some cases the results are quite the opposite.

See:

Pană, M–C & Fanea-Ivanovici, M 2019, ‘Institutional Arrangements and Overeducation: Challenges for Sustainable Growth. Evidence from the Romanian Labour Market’, Sustainability, Vol. 11, pp. 1-19, doi:10.3390/su11226459.

Ortiz, L. Not the right job, but a secure one: Over-education and temporary employment in France, Italy and Spain. Work Employ. Soc. 2010, 24, 47–64.

Pearlman, S.; Rubb, S. The impact of education-occupation mismatches on wages in Mexico. Appl. Econ. Lett. in press.

Dolton, P.; Vignoles, A. The incidence and effects of overeducation in the UK graduate labour market. Econ. Educ. Rev. 2000, 19, 179–198.

Wu, N.; Wang, Q.Y. Wage penalty of overeducation: New micro-evidence from China. China Econ. Rev. 2018, 50, 206–217.

Reviewer 2 Report

From a methods and analysis perspective this research is sound. On the other hand, I have suggestions to make the paper more readable as well as conform with format/presentation. I suggest professional editing

Lines 68 – 83 should be consolidated into one more precise paragraph.

The “Theoretical Framework” is quite interesting; Figure one should be the focus and explain the rational for the direct and indirect effects. Much in this section is literature review and explanations are not concise. Four or five pages devoted to the theoretical framework is not in keeping with most articles.

Study Design – Presentation needs serious revision to improve the readability. Tables 1, 2 and 3 present relevant information is just one example. My suggestion is to place the several tables I n an appendix or make them available to researchers, beyond this, results, are presented – Tables 6 – 8 – which should appear in the Results Section.

The paper mixes Results and Analysis/Interpretation together. Succinctly present the statistical analysis in a separate section and analysis of the results in separate sections. Considerable editing is necessary.

Upon reading the Conclusions Recommendations, again they need to be more succinct.

I notice there is no section on limitations of the study. A minor point: Abstract should mention sample size; did the authors compare characteristics of the sample used in the study for analysis to those who were excluded?

Reviewer 3 Report

The topic of the paper is interesting and important. It is well motivated in the introduction and the methods used seem adequate.

The results obtained in the empirical section are interesting and useful . However, from my point of view, the discussion could be much stronger with a closer link with previous studies on related issues.The revision of this section is my main suggestion.

After a careful revision of that section, I think the paper could be considered for publication.

Reviewer 4 Report

The authors have:

1. Chosen a hot topic that deserves timely research, discussion, and follow-up;

2. Demonstrated sufficient analysis and discussion of previous studies in the literature review;

3. Employed stratified random sampling with data collected from various schools,

5. Conducted thorough statistical analyses and explained results in sufficient detail; &

6. Provided practical implications.

Minor Revisions are required. Please address the following comments and questions.

1.      Pls assign separate sections for research gaps and objectives respectively

2.      Elaborate on the research gaps

3.      Incorporate “limitations” and “future research directions” sections at the end of the paper and elaborate

4.      Supplement with validity and reliability statistics for the questionnaire scales

5.      Suggest to remove numbering within paragraphs -> i.e.,

6.      Table 8:  replace “NO” with “No”

Round 2

Reviewer 1 Report

The authors made improvements to this article by emphasizing the importance of institutional arrangements concerning entrepreneurial behavior. However, the analysis can still be improved in the particular matter of setting the theoretical framework and the purpose of this study.

Some of the authors’ arguments are presented above followed by, hopefully, useful comments and reference suggestions.

”Based on the above research background, this paper takes the university town of Jinan City, Shandong Province as a case area, and will systematically study the relationship between college students' non-cognitive ability and entrepreneurial intention from a multi-dimensional perspective, as well as the mediating role of social support perception in the relationship between the two, with a view to fundamentally understand the factors and mechanisms that affect college students' entrepreneurial willingness, and provide entrepreneurial guidance and services to college students in a targeted manner. At the same time, it provides decision-making reference for promoting social harmony, stability, economic health and sustainable development.”

The authors slightly changed the aim of this study, without altering its main purpose, in order to narrow down the research area they had previously announced. However, maintaining on the need to “fundamentally understand the factors and mechanisms that affect college students' entrepreneurial willingness” one cannot avoid the importance of institutions. Moreover, the authors emphasized the importance of institutional arrangements. Institutional arrangements cannot exist without institutions and their fundamental role for the economic and social activities. Therefore, understanding the internal and external factors (formal and informal institutions) that affect students’ willingness to become entrepreneurs, as the purpose of this study, is a fundamental need.

See:

North, D.C. Understanding the Process of Economic Change; Princeton University Press: Princeton, NJ, USA, 2005; pp. 103–115. ISBN 0-691-11805-1.

Williamson, O.E. The New Institutional Economics: Taking Stock, Looking Ahead. J. Econ. Lit. 2000, 38, 595–613.

Sobel, R.S. Testing Baumol: Institutional quality and the productivity of entrepreneurship. J. Bus. Ventur. 2008, 23, 641–655.

Boettke, P.J.; Fink, A. Institutions first. J. Econ. 2011, 7, 499–504.

6.2 Colleges and universities should improve the quality of entrepreneurship education and promote entrepreneurial actions

“[...] the government has promulgated many policies to support college students' entrepreneurship, reflecting the government's strong support for the use of entrepreneurship to promote employment.”

If institutional environment matters, as the authors also emphasized, one cannot take for granted government intervention just because the purpose is supposedly beneficial. In addition, government intervention means changes in the institutional framework that affects entrepreneurial initiatives. Such changes may create benefits, but also losses to the becoming and the already existing entrepreneurs. Therefore, the recommendations that the authors made should not be taken as granted. Moreover, there are studies that stress the importance of analyzing the outcomes of such “policies” concerning entrepreneurship and employment to which the authors of this manuscript should refer.

See:

Pană, M–C & Fanea-Ivanovici, M 2019, ‘Institutional Arrangements and Overeducation: Challenges for Sustainable Growth. Evidence from the Romanian Labour Market’, Sustainability, Vol. 11, pp. 1-19, doi:10.3390/su11226459.

Turmo-Garuz, J.; Bartual-Figueras, M.T.; Sierra-Martinez, F.J. Factors Associated with Over-Education Among Recent Graduates. Soc. Indic. Res. 2019, 144, 1273–1301

Kucel, A.; Vilalta-Bufi, M. University Program Characteristics and Education-Job Mismatch. BE J. Econ. Anal. Policy 2019, 19, 20190083.

Ortiz, L. Not the right job, but a secure one: Over-education and temporary employment in France, Italy and Spain. Work Employ. Soc. 2010, 24, 47–64.

Reviewer 2 Report

The authors are commended for making significant revisions to the manuscript.  The readability is improved somewhat, but not as nearly as much as I had hoped.
